# Self-rated health and perceived environmental quality in Brunei Darussalam: a cross-sectional study

Evi Nurvidya Arifin ,[1] Chang-Yau Hoon,[1] Ly Slesman,[1] Abby Tan[2]

[1]Centre for Advanced Research (CARe), Universiti Brunei Darussalam, Gadong, Brunei Darussalam
[2]Chancellery, Universiti Brunei Darussalam, Gadong, Brunei Darussalam

**Correspondence to**
Dr Evi Nurvidya Arifin;
evi.arifin@ubd.edu.bn

## ABSTRACT

**Objectives** This paper examines the relationship between individuals' perceptions of environmental quality and self-rated health (SRH) after controlling for dimensions of socioeconomic, demographic and healthy lifestyle variables.

**Design** A cross-sectional survey.

**Setting** The survey was conducted in Belait, an oil-rich and gas-rich district in Brunei Darussalam, from 17 October to 11 November 2019 and focused on the most populated subdistricts (Kuala Belait, Seria and Liang), where 97% of the people reside.

**Participants** A final sample of 1000 respondents aged 18 years and older were randomly selected from the population of the chosen subdistricts, with 95% CI and ±3 margin of error. Due to variable selection, only 673 respondents were available for analysis.

**Outcome measures** SRH was dichotomised into 1 for good health and 0 otherwise. Perceptions of environmental quality included perceptions of the natural environment (air quality, marine quality, water supply, noise and olfactory pollution) and the social environment (crime). $\chi^2$ and logistic regression models were used to assess the relationship between individuals' perceived environmental quality and SRH.

**Results** Most respondents perceived themselves with good SRH (72%). The adjusted logistic regression shows that perceptions of air quality (OR=2.20, 95% CI 1.15 to 4.22, p=0.018) and marine resources (OR=1.84, 95% CI 1.24 to 2.74, p=0.002) in their surrounding areas were significantly associated with good SRH. However, other environmental variables were insignificantly associated with SRH. Among the control variables, healthy lifestyle and employment had positive associations with good SRH (OR=3.89, 95% CI 1.96 to 7.71, p=0.000, for exercising 3–5 times a week; OR=1.72, 95% CI 1.09 to 2.71, p=0.021, for being employed). In addition, frequent physical exercise compensated for the negative health impact of environmental pollution.

**Conclusions** This study suggests that environmental quality has an important role in SRH. However, a healthy lifestyle measured with frequency of physical exercise seems to compensate for the adverse environmental effects on SRH.

## STRENGTHS AND LIMITATIONS OF THIS STUDY

⇒ The strength of this study is in providing new empirical evidence on how environmental quality is related to self-rated health, a subjective measure, especially in the oil-rich and gas-rich Belait District of Brunei Darussalam, one of the Southeast Asian countries.

⇒ Another strength is that the study used a representative and relatively large district sample.

⇒ One limitation is that there is no comparable measure of objective health.

⇒ Perceptions of olfactory and noise pollutants were grouped into one category and there was only one variable for social environment; further studies should therefore collect data on these pollutants separately and more variables for social environment.

⇒ A longitudinal study is important to further examine the causal relationship between perceptions of environmental quality and health and to investigate the dynamics of the relationship.

has far-reaching consequences to individuals, families, communities and states. At the individual level, being unhealthy encompasses suffering, deprivation of self-confidence and capability to function, and losing productivity and various opportunities. To some extent, being unhealthy can lead to a premature end of life. What happens to individuals may impact the whole family. Unhealthy or sick family members may lead to a family caregiving crisis and disrupt other family members' social and economic activities. They may even fall into poverty.[2–4] Communicable diseases, for example, may even become a pandemic, resulting in sudden considerable disruption in people's life. At the state level, a health crisis can weaken the healthcare system and destabilise the economy through contraction in employment, increased state expenditure and reduced investment. The current COVID-19 pandemic provided evidence of the impact of a health crisis on both health and economy in many countries around the globe.[5–7]

## INTRODUCTION

Ensuring healthy life is goal 3 of the 17 sustainable development goals.[1] Health crisis

To be healthy has multidimensional factors,[8] with complex roles at different levels (individual, families, community and state). Previous studies were mostly concerned with health-related individual and family factors such as medical, biological or hereditary conditions; psychosocial, demographic, social and economic conditions; and healthy lifestyle (HLS).[9–14] State-level health status factors were only considered when the analyses were concerned with cross-country differences. For example, the level of democratic governance was found to have a positive association with health.[15] Yet studies within a country or region should also consider health-related community factors, especially the importance of the surrounding environmental condition. This environmental condition can take three forms: built or man-made, natural and social environment.[16]

Because it is not easy to measure objective measures of environmental quality at the individual level, this paper used its proxy—perceived environmental quality. Most importantly, objective measures of environmental quality apply to all individuals living in the same environment and there is no variation among individuals. Thus, the environment–health relationship does not exist in a small study area. The use of subjective measures of environmental quality can vary from one individual to another. Perceptions of environmental quality may matter to individuals. Individuals' perceptions of environmental quality may say more about variations in the perceptions of environment and health. This study contributes to new empirical findings. This subjective variable was then used to examine the association between the environmental quality of the surrounding areas and self-rated health status. Similarly, in population-based surveys, health status was assessed subjectively, known as self-rated health (SRH), which has been widely used in many countries. SRH is a reliable health indicator used in population-based studies[17 18] and provides a reliable and acceptable approximation of the overall health outcomes of a population.[18–20] It captures the biological, mental and social states of an individual's health[21] and has been shown to explain mortality and morbidity[21 22] and the causes of deaths.[14 23]

Assessment of the association between these two subjective domains has been conducted mostly in Western countries,[24–30] as well as in some Asian countries such as Sri Lanka,[31] China,[16 32] Japan and South Korea.[33] However, little is known about this relationship in Southeast Asian countries, where countries have prospered and developed significantly. Natural environmental issues in this region, such as seasonal forest fires, have also been a transboundary concern.[34]

Environment is a broad concept and has been measured differently from one study to another. For example, a study in England and Scotland[24] measured neighbourhood environmental conditions related to man-made and social environments, such as access to private transportation, physical quality of the residential environment, political climate and political engagement. Similarly, another study in the UK[28] used these

environmental conditions but the selected variables referred to access to amenities, neighbourhood quality and neighbourhood disorder. Unlike these studies,[24 28] the environmental condition of the residents of Chelsea, Massachusetts in the USA was measured differently by asking about their perceptions of the natural environment with regard to air quality, odour and noise, and of the social environment including feeling safe, neighbourhood crime and social cohesion.[26] A study in Sri Lanka[31] examined environmental factors using ventilation problems, water shortage, garbage disposal problems, mosquito threat, stray dogs and social environment (nuisance from neighbours and overcrowding). A study in China concerning its rapid development examined people's perceptions of air pollution, water pollution, freshwater resource shortage and green space shortage in relation to SRH.[16]

However, findings from the abovementioned studies were inconclusive. The study in England and Scotland found that most of their selected neighbourhood environmental factors did not associate with SRH.[24] Similarly, none of the environmental factors was associated with SRH.[31] In contrast, some studies found that people's poor living environment was likely associated with a higher likelihood of reporting poor SRH.[25–29 33] Urban residents in China who perceived air pollution and noise pollution were more likely to report not in good SRH.[32] Air pollution has an impact on respiratory diseases in some Southeast Asian countries[35] and on non-communicable diseases such as chronic kidney disease in 194 countries and territories.[36] Different results were also found in China,[32] which perceived environmental pollution was not significantly associated with SRH among the country's rural residents. Nevertheless, there has been an insufficient number of empirical studies that assessed the impact of perceived environmental quality on SRH among Southeast Asian countries.

With this background, identifying environmental factors of health is crucial to contribute new empirical findings to the debate on the environment–health relationship, more specifically between perceived environmental quality and SRH. A study in China[16] reported that an advantage of examining subjective indicators compared with objective indicators of the environment–health relationship is providing more stories about an individual's life. This study aims to fill the gap in this relationship by controlling for demographic, socioeconomic (SES) and lifestyle variables in Brunei Darussalam (hereafter, Brunei). This study does not consider variables measuring built environmental quality[16] due to unavailability of the information. Our primary objectives are therefore to estimate the prevalence of good SRH and to examine the relationship between natural and social environmental qualities and SRH, accounting for dimensions of SES, demographic and HLS variables based on a cross-sectional study. Brunei's development has led to a rapid urbanisation, transforming its green nature and social environment for human needs.

Brunei has also provided its people with high quality of life, as indicated by its high Human Development Index, which stood at the 43rd position worldwide in 2018.[37] Its economy is heavily dependent on oil and gas production, with oil and natural gas as its major exports and its main source of development and wealth since the mid-20th century. Brunei benefits considerably from the world market oil price, making it the country with the second highest per capita gross domestic product (GDP), after Singapore, among the Southeast Asian countries.

## METHODS
### Design, setting, study size and participants
We used the Strengthening the Reporting of Observational Studies in Epidemiology cross-sectional checklist when writing our report.[38] The paper used an original cross-sectional survey conducted by Universiti Brunei Darussalam between 17 October and 11 November 2019, before the COVID-19 pandemic spread to this country in March 2020. The survey was conducted in Belait District, the largest of the four districts in Brunei in terms of land size. This district was home to around 75 900 persons, or 16.5% of Brunei's population of 459 500, in 2019.[39] Within this district, three subdistricts were selected, namely Seria, Liang and Kuala Belait, where the majority of people (97%) reside and where the primary sources of the country's economy—the production of hydrocarbon resources—are located. They are located in the northern coastal area, facing the South China Sea, as seen in figure 1.

This survey interviewed 1000 randomly selected respondents aged 18 years and above residing in the chosen subdistrict (or *mukim*). Based on the 2016 population census, the population aged 18 years and above in Belait was roughly 72%.[40] This sample size was randomly drawn with a 95% confidence level and ±3% margin of error. Face-to-face interviews were performed by 11 trained research assistants. The survey gathered comprehensive information using a pretested structured questionnaire covering background characteristics, employment and income, housing, education and aspirations, health, and

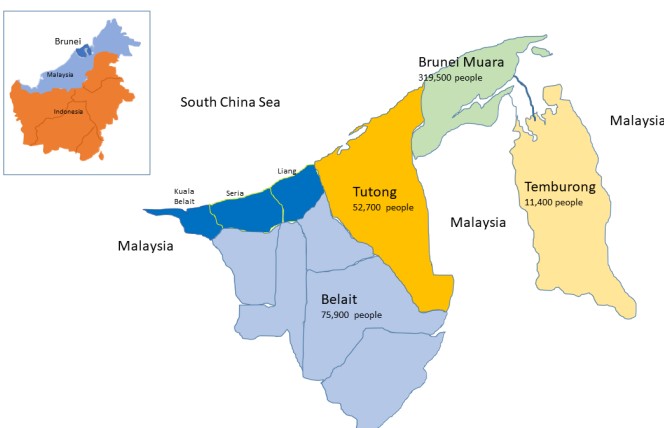

**Figure 1** Geographical location of the studied areas.

environmental impact and awareness. The questions were mostly closed-ended, combined with several open-ended ones. Prior to the interviews, respondents were aware and informed about the survey's objectives. The response for each selected variable varied for various reasons, such as do not want to answer, do not know, unclear reason and others. For consistency, 673 respondents who completely responded to all questions were selected for this paper. Table 1 provides further details on the basic characteristics of the sample.

### Patient and public involvement
Patients and/or the public were not involved in the design, or conduct, or reporting, or dissemination plans of this research.

### Variables and measurements
#### Dependent variable: SRH
SRH has been widely used in population-based surveys due to its simplicity as a single question. The question usually takes forms such as 'In general, would you say that your health is …?'[41–43] and a Likert scale is usually used for the response. However, in practice, the question varies slightly in different contexts. For instance, the question 'Would you say your overall physical health is…?' was used by a study in the USA.[20] Studies in Japan and Korea used the question 'How would you rate your health?'[17] and in Norway 'How is your health at the moment'.[44] In Timor Leste and India, self-rated physical health was measured by more than one variable and therefore expressed as an index.[43 45] Most of these studies used a Likert scale.

SRH in this paper was assessed with a single question: 'Please select the item that accurately describes your overall current health'. The response item used a 5-point Likert scale (very good, good, fair, poor, very poor). Guided by common practice,[16 20 24 26 46–48] this paper classified the responses into a dichotomised dependent variable to measure good SRH by combining very good and good health into 1 and 0 otherwise.

#### Focal variables: perceived environmental quality
Perceptions of environmental quality (*PEQ*), the main independent variable, consisted of natural and social environmental quality. Data on natural environmental quality were collected by asking respondents to state the extent of pollution in terms of air quality, marine resources quality, and other pollutions such as noise, olfactory/odour and others; quality of social environment was indicated by neighbourhood crime. These variables were measured at different points on a Likert scale. The question 'How would you characterize the air quality in your area?' had three scales (1=very clean, 2=fairly clean and 3=not clean); the last category was treated as the reference group in the logistic model. Two other questions were asked on the importance of air quality improvement and the commitment to it: 'How important is air quality improvement to you?' and 'Compared to other people,

**Table 1** Respondent characteristics and comparison of good SRH

| Variable | | Frequency | Per cent | Good SRH | $\chi^2$ value (p value) |
|---|---|---|---|---|---|
| Total | | 673 | 100.0 | – | |
| SRH | Not good | 184 | 27.3 | – | |
| | Good health | 489 | 72.7 | – | |
| Sex | Female | 352 | 52.3 | 72.4 | 0.002 (0.964) |
| | Male | 321 | 47.7 | 72.9 | |
| Age | 18–29 | 241 | 35.8 | 77.2 | 7.785 (0.020) |
| | 30–59 | 343 | 51.0 | 72.3 | |
| | 60–85 | 89 | 13.2 | 61.8 | |
| Marital status | Married | 374 | 55.6 | 71.9 | 2.453 (0.293) |
| | Ever married | 45 | 6.7 | 64.4 | |
| | Single | 254 | 37.7 | 75.2 | |
| Ethnic group | Brunei Malay | 426 | 63.3 | 74.9 | 3.292 (0.349) |
| | Indigenous Malay | 116 | 17.2 | 70.7 | |
| | Chinese | 75 | 11.1 | 66.7 | |
| | Others | 56 | 8.3 | 67.9 | |
| Education | Primary and below | 37 | 5.5 | 54.1 | 6.926 (0.074) |
| | Secondary | 242 | 36.0 | 74.0 | |
| | Postsecondary and diploma | 221 | 32.8 | 74.2 | |
| | High degree | 173 | 25.7 | 72.8 | |
| House ownership | Owned house | 304 | 45.2 | 70.1 | 1.878 (0.171) |
| | Not owned | 369 | 54.8 | 74.8 | |
| Employment status | Employed | 437 | 64.9 | 76.4 | 12.578 (0.002) |
| | Unemployed | 44 | 6.5 | 77.3 | |
| | Not labour force | 192 | 28.5 | 63.0 | |
| Physical exercise | Never | 142 | 21.1 | 63.4 | 22.956 (0.000) |
| | 1–2 times | 362 | 53.8 | 70.4 | |
| | 3–5 times | 128 | 19.0 | 88.3 | |
| | Every day | 41 | 6.1 | 75.6 | |

Source: Authors' calculation based on data collected for the study.
SRH, self-rated health.

how would you describe your commitment to preserving and improving air quality?' Improvement in air quality was treated as a dichotomous variable, with the value of 1 for very important or somewhat important and 0 for not very important or not important at all. Commitment to improve air quality was treated as a dichotomous variable, with the value of 1 for much more committed, somewhat more committed or about the same, and 0 for somewhat less committed or much less committed.

Perceptions of water supply quality and other pollution were determined by a yes or no answer to the following questions: 'Do you have any problem with water supply?' and 'Is the dwelling exposed to noise, odour, or pollution problems?' A 'yes' answer was given the value of 1 and 0 otherwise. The response to the following question was treated as a dichotomous variable, with a value of 1 for very good, good, or not good not bad, and 0 for bad or very bad: 'How would you

describe the condition of the marine resources (eg, fisheries, coral reefs etc.) in your area today?' Only one variable was available for quality of the social environment: 'Do you consider your area a high, medium or low crime area?' It was treated as a dichotomous variable, with 1 for medium or high crime and 0 for low crime.

### Control variables
#### Demographic and SES variables
Control variables consisted of demographic, SES and HLS variables. Demographic variables included age, sex, marital status and ethnic group. Sex was treated as a dichotomous variable, with the value of 1 for male and 0 for female. Age was treated as a numerical variable for the logistic model and a categorical variable (18–29, 30–59, and 60–85 years old) for $\chi^2$ analysis. Marital

status was categorised into three groups: married, ever married (divorced and widowed) and single/never married, with married as the reference group. Ethnicity consisted of four groups: Brunei Malay, Indigenous Malay, Chinese and others. As of the Brunei Nationality Act 1961, the Malay group consists of seven ethnic groups: Brunei Malay or Brunei, Belait, Bisaya, Dusun, Kedayan, Murut and Tutong.[49] Brunei Malay is culturally Malay and embraces Islam, similar to the Malay in West Malaysia, while other Indigenous Malays are culturally and religiously diverse.[50]

Socioeconomic variables (*SES*) included education, employment status and ownership of the house. Education had four groups: primary, secondary, postsecondary and higher degree. Employment status was categorised into employed, unemployed and not in the labour force (ie, those who are neither in employment nor in unemployment). Being not in the labour force was treated as the reference group. House ownership was a dichotomous variable, with 1 for owning a house and 0 for not owning a house.

### Healthy lifestyle

Due to limited information, a healthy lifestyle (*HLS*) was measured with a survey question asking about the frequency of doing physical activities: 'How often do you play sports or exercise per week?' It had four responses: 1=never, 2=one to two times, 3=three to five times and 4=every day. Never playing sport or doing exercise was the reference category.

### Statistical methods

$\chi^2$ statistics were initially performed through a series of cross-tabulations between SRH status and all selected variables. Logistic regression model[51] was used to assess the association between perceived environmental quality and SRH. The following is the logistic regression model used in this paper.

$$\ln\left(\frac{p}{1-p}\right) = \beta_0 + \beta_j PEQ_j + \beta_k DEM_k + \beta_l SES_l + \beta_m HLS_m + \varepsilon$$

where *p* refers to the probability of good SRH; *PEQ* refers to perceived environmental quality, with *j* indicators; *DEM* refers to demographic control variable, with *k* indicators; *SES* refers to socioeconomic variables, with *i* indicators; and *HLS* refers to a healthy lifestyle, with only one indicator.

Following a previous study,[17] the model was applied in few stages to examine each group of variables at each stage. This paper has four stages to observe the possible changes played by perceived environmental quality, demographic variables, SES variables and lifestyle variables in SRH. Model 1 assessed the association between SRH and PEQ without control variables. Model 2 enriched model 1 by adding demographic variables. Model 3 further added SES, while model 4 added HLS as controls. P=0.05 was selected to determine the significance of the variables. OR and its 95% CI were provided to examine the significance of the relationship.

## RESULTS

### Sample description

An overview of the characteristics of the respondents based on the demographic, SES and HLS variables is presented in table 1. More than half were female (52.3%) and of mature ages between 30 and 59 years old (51.0%). Majority were married (55.6%), with a significant percentage of single individuals (37.7%). The rest were ever married, either widowed or divorced (6.7%). Brunei Malay accounted for the largest ethnic group (63.3%). More than half attended postsecondary education and above (58.5%) and about 65% were employed. Less than half owned the house they lived in. Regarding HLS, only one-fifth never participated in physical exercise and many of them had physical activities once or twice per week.

### Perceived environmental quality

Analysis of the survey data shows a good perception of air quality in the area where the respondents reside. The descriptive statistics show that 80.2% of the respondents perceived that they reside in areas with fairly good air quality. Only 12.5% of the respondents perceived they live in an area with very good air quality. In other words, only a small percentage of respondents (7.3%) perceived they live in air-polluted areas. They expressed the need to improve the air quality and half of them were committed to it (table 2). About 77% did not have problems with sources of water supply in their daily life. Majority (73.7%) did not perceive the existence of other pollutions such as noise, olfactory and others (table 2). However, a lower percentage, more than half of them (53.8%), perceived that the condition of marine resources in the district was not good. The social environmental quality was reported frequently by 71.9% of the residents to have low crime.

### Differential of SRH: $\chi^2$ analysis

Overall, the majority (72.7%) perceived good SRH. The $\chi^2$ analysis presented in table 2 shows that the prevalence of good SRH significantly differed by perceived air quality ($\chi^2$=12.9, p=0.002), marine resources ($\chi^2$=10.3, p=0.001) and water supply ($\chi^2$=5.4, p=0.026). The prevalence of good SRH among respondents living in areas perceived to have very clean air was higher (77.4%) than those living in areas with fairly clean air (73.9%) and much higher than those living in areas with polluted or not clean air (52.0%). Perceived air quality was thus positively associated with good SRH. However, the prevalence of good SRH was not significantly different between respondents who perceived it is very important to improve air quality and those who perceived it is not important to do so (table 2). In other words, commitment to improving air quality was not associated with good SRH, although perceived air quality was positively related to good SRH. This indicates the existence of a knowledge–attitude gap.

The prevalence of good SRH was higher among those living in areas perceived to have good marine resources than those living in areas not having good marine resources (78.8% and 67.4%, respectively, p<0.001). The

**Table 2** Differential of good SRH by perceived environmental quality

| | | Frequency | Per cent | Good SRH | $\chi^2$ value (p value) |
|---|---|---|---|---|---|
| Air quality | Very clean | 84 | 12.5 | 77.4 | |
| | Fairly clean | 540 | 80.2 | 73.9 | 12.903 (0.002) |
| | Not clean | 49 | 7.3 | 51.0 | |
| Water problem | No problem | 517 | 76.8 | 74.9 | 5.410 (0.026) |
| | Water supply problem | 156 | 23.2 | 65.4 | |
| Marine | Not good | 362 | 53.8 | 67.4 | 10.330 (0.001) |
| | Good | 311 | 46.2 | 78.8 | |
| Other pollutions | Not exposed | 496 | 73.7 | 74.0 | 1.440 (0.230) |
| | Exposed to noise, odour and pollution problems | 177 | 26.3 | 68.9 | |
| Crime | Low crime | 484 | 71.9 | 73.3 | 0.296 (0.586) |
| | Medium and high crime | 189 | 28.1 | 70.9 | |
| Air quality improvement | Not very/not important | 169 | 25.1 | 71.6 | 0.067 (0.796) |
| | Important to improve air quality | 504 | 74.9 | 73.0 | |
| Commitment | Less committed | 313 | 46.5 | 72.2 | 0.026 (0.873) |
| | Committed | 360 | 53.5 | 73.1 | |

Source: Authors' calculation based on data collected for the study.
SRH, self-rated health.

prevalence of good SRH was different between respondents with no exposure to noise, olfactory and other pollutants and those exposed to these pollutants (74.0% and 68.9%, respectively), although the $\chi^2$ result shows no significant difference ($\chi^2=1.4$, p=0.230).

The prevalence of good SRH was higher among those living in a house without any problems with water supply (74.9%) than those having problems with water supply (65.4%, p<0.005). Social environmental condition, measured with perceptions of safety from crime, was not associated with good SRH.

Age mattered in health status. Table 1 shows that the prevalence of good SRH was lower among those of older age than those of younger age. Men and women perceived almost equally good SRH (72.9% and 72.4%, respectively; $\chi^2=0.002$, p=0.964). There were also no ethnic differentials in the prevalence of good SRH ($\chi^2=3.3$, p=0.349).

Among the SES variables, employment status is the only variable which significantly differentiated good SRH ($\chi^2=12.6$, p=0.002). The prevalence of good SRH among those who were either employed (76.4%) or unemployed (77.3%) was higher than among those who were not in the labour force (63.0%). Marital status, education and home ownership were not significantly associated with SRH.

Physical exercise mattered more ($\chi^2=23.0$, p=0.000). The prevalence of good SRH was much higher (88.3%) among respondents exercising regularly between three and five times per week than those who never did any physical exercise (63.4%).

### Logistic regression of SRH

There are four logistic regression results presented in tables 3 and 4. Model 1 examines the association between PEQ and SRH without controlling for other sets of variables. This model shows that respondents with perceptions of living in areas with fairly clean air (OR=2.42, 95% CI 1.30 to 4.49, p=0.005) or areas with very clean air (OR=2.40, 95% CI 1.07 to 5.38, p=0.034) were more likely to state good SRH than those living in areas with air pollution. This pattern was slightly different from the $\chi^2$ results, with respondents in areas with very clean air having the highest prevalence of good SRH. While a further examination (not shown in the table) indicates that the difference between these two groups was insignificant, the association between perceptions of air quality and good SRH remains significant.

Respondents' opinion and commitment to improving air quality were not associated with SRH (table 3). Respondents living in areas with perceived good marine resources were more likely to report good SRH than those living in areas with marine resources that were not in good condition (OR=1.69, 95% CI 1.18 to 2.44, p=0.005).

However, good health was not significantly associated with respondents' perceptions of other environmental qualities (water supply, and noise, olfactory and other pollution). Model 1 changes the significance level of the association between water supply and good SRH into insignificant from the $\chi^2$ results. This indicates that the presence of perceptions of air quality and marine had a more salient influence on the association. Neither was

**Table 3** OR for good SRH and perceived environmental quality controlled for demographic variables

| Variables | Model 1 | | | Model 2 | | |
|---|---|---|---|---|---|---|
| | OR | 95% CI | Significance | OR | 95% CI | Significance |
| Perceived environmental quality | | | | | | |
| Air quality (ref: not clean) | | | 0.020 | | | 0.024 |
|    Very clean | 2.40 | 1.07 to 5.38 | 0.034 | 2.37 | 1.04 to 5.41 | 0.040 |
|    Fairly clean | 2.42 | 1.30 to 4.49 | 0.005 | 2.39 | 1.28 to 4.49 | 0.007 |
| Marine resources (ref: not good) | | | | | | |
|    Good marine resources | 1.69 | 1.18 to 2.44 | 0.005 | 1.78 | 1.22 to 2.60 | 0.003 |
| Other pollutions (ref: not exposed) | | | | | | |
|    Exposed to noise, odour and others | 1.03 | 0.68 to 1.55 | 0.907 | 1.02 | 0.68 to 1.56 | 0.908 |
| Water supply (ref: no problem) | | | | | | |
|    Problem with water supply | 0.71 | 0.47 to 1.08 | 0.110 | 0.74 | 0.49 to 1.13 | 0.166 |
| Crime (ref: low crime) | | | | | | |
|    Medium and high crime | 0.98 | 0.67 to 1.45 | 0.933 | 0.90 | 0.61 to 1.34 | 0.610 |
| Air quality improvement (ref: not important) | | | | | | |
|    Important | 0.96 | 0.64 to 1.45 | 0.857 | 0.96 | 0.63 to 1.45 | 0.830 |
| Commitment to air quality improvement (ref: not committed) | | | | | | |
|    Committed | 1.08 | 0.75 to 1.54 | 0.687 | 1.02 | 0.71 to 1.46 | 0.934 |
| Demographic variables | | | | | | |
| Age (in years) | – | – | – | 0.98 | 0.96 to 0.99 | 0.001 |
| Sex (ref: female) | | | | | | |
|    Male | – | – | – | 1.02 | 0.71 to 1.46 | 0.913 |
| Ethnic group (ref: Brunei Malay) | | | | | | 0.588 |
|    Indigenous Malay | – | – | – | 0.82 | 0.51 to 1.31 | 0.397 |
|    Chinese | – | – | – | 0.86 | 0.49 to 1.51 | 0.597 |
|    Others | – | – | – | 0.68 | 0.37 to 1.28 | 0.231 |
| Marital status (ref: married) | | | | | | 0.409 |
|    Ever married | – | – | – | 0.86 | 0.42 to 1.74 | 0.669 |
|    Single | – | – | – | 0.72 | 0.44 to 1.19 | 0.201 |
|    Constant | 1.02 | | 0.967 | 3.70 | | 0.018 |

Source: Authors' calculation based on data collected for the study.
ref, reference group; SRH, self-rated health.

the social environmental condition measured by perceptions of crime associated with SRH. These findings on the association between perceived environmental quality and good SRH remain the same after controlling for demographic variables (model 2).

It is worth briefly discussing the findings on demographic variables. Among the selected demographic variables, age was negatively associated with SRH (model 2). The odds of stating good SRH decreased with age (OR=0.98, 95% CI 0.96 to 0.99, p=0.001). This is in contrast to the relationship between gender and good SRH, with OR=1 and p=0.913. In other words, men and women had the same likelihood of reporting good SRH. There was also no difference in the likelihood of reporting good SRH in terms of marital status and ethnic group. This deserves further study.

Adding SES variables in model 3 (table 4), all associations between perceived environmental quality and SRH remain the same, with a slight change in the magnitude of perceived air quality and marine resources. The adjusted ORs of reporting good SRH were lower in areas with fairly clean air (OR=2.35, 95% CI 1.25 to 4.44, p=0.008) and higher in areas with very clean air (OR=2.44, 95% CI 1.06 to 5.63, p=0.036) than the ORs presented in the unadjusted model (model 1). In other words, model 3 shows a positive relationship between perceived air quality and good SRH, which was similar to the $\chi^2$ results.

**Table 4** OR for good SRH and perceived environmental quality controlled for demographic, socioeconomic and healthy lifestyle variables

| Variables | Model 3 | | | Model 4 | | |
|---|---|---|---|---|---|---|
| | OR | 95% CI | Significance | OR | 95% CI | Significance |
| Perceived environmental quality | | | | | | |
| Air quality (ref: not clean) | | | 0.029 | | | 0.059 |
| Very clean | 2.44 | 1.06 to 5.63 | 0.036 | 2.19 | 0.93 to 5.12 | 0.072 |
| Fairly clean | 2.35 | 1.25 to 4.44 | 0.008 | 2.20 | 1.15 to 4.22 | 0.018 |
| Marine resources (ref: not good) | | | | | | |
| Good marine resources | 1.77 | 1.20 to 2.60 | 0.004 | 1.84 | 1.24 to 2.74 | 0.002 |
| Other pollutions (ref: not exposed) | | | | | | |
| Exposed to noise, odour and others | 1.07 | 0.70 to 1.63 | 0.771 | 0.98 | 0.63 to 1.51 | 0.911 |
| Water supply (ref: no problem) | | | | | | |
| Problem with water supply | 0.72 | 0.47 to 1.11 | 0.136 | 0.71 | 0.46 to 1.10 | 0.123 |
| Crime (ref: low crime) | | | | | | |
| Medium and high crime | 0.90 | 0.60 to 1.34 | 0.595 | 0.97 | 0.64 to 1.45 | 0.866 |
| Air quality improvement (ref: not important) | | | | | | |
| Important | 0.97 | 0.64 to 1.47 | 0.874 | 0.98 | 0.64 to 1.50 | 0.926 |
| Commitment to air quality improvement (ref: not committed) | | | | | | |
| Committed | 0.99 | 0.69 to 1.44 | 0.975 | 0.92 | 0.63 to 1.35 | 0.681 |
| Demographic variables | | | | | | |
| Age (in years) | 0.98 | 0.96 to 0.99 | 0.033 | 0.98 | 0.97 to 1.00 | 0.079 |
| Sex (ref: female) | | | | | | |
| Male | 0.98 | 0.68 to 1.42 | 0.912 | 0.86 | 0.59 to 1.26 | 0.433 |
| Ethnic group (ref: Brunei Malay) | | | 0.599 | | | 0.384 |
| Indigenous Malay | 0.78 | 0.49 to 1.26 | 0.314 | 0.78 | 0.48 to 1.26 | 0.303 |
| Chinese | 0.87 | 0.49 to 1.54 | 0.632 | 0.83 | 0.46 to 1.48 | 0.519 |
| Others | 0.71 | 0.37 to 1.34 | 0.287 | 0.60 | 0.31 to 1.16 | 0.126 |
| Marital status (ref: married) | – | | 0.585 | – | | 0.552 |
| Ever married | 0.89 | 0.42 to 1.86 | 0.747 | 0.77 | 0.36 to 1.63 | 0.487 |
| Single | 0.77 | 0.46 to 1.29 | 0.320 | 0.79 | 0.47 to 1.34 | 0.384 |
| Socioeconomic variables | | | | | | |
| Education (ref: primary and below) | | | 0.550 | | | 0.435 |
| Secondary | 1.35 | 0.60 to 2.99 | 0.468 | 1.27 | 0.56 to 2.88 | 0.567 |
| Postsecondary and diploma | 1.04 | 0.44 to 2.46 | 0.934 | 0.93 | 0.39 to 2.26 | 0.876 |
| Higher education | 0.99 | 0.41 to 2.38 | 0.981 | 0.87 | 0.35 to 2.14 | 0.755 |
| Employment (ref: not labour force) | | | 0.091 | | | 0.068 |
| Employed | 1.63 | 1.05 to 2.53 | 0.029 | 1.72 | 1.09 to 2.71 | 0.021 |
| Unemployed | 1.53 | 0.66 to 3.57 | 0.323 | 1.63 | 0.68 to 3.89 | 0.270 |
| House ownership (ref: not owned) | | | | | | |
| Owned | 1.09 | 0.73 to 1.63 | 0.669 | 1.11 | 0.60 to 1.36 | 0.626 |
| Healthy lifestyle | | | | | | 0.000 |
| Physical activities (ref: never) | | | | | | |
| 1–2 times | – | – | – | 1.06 | 0.67 to 1.68 | 0.795 |
| 3–5 times | – | – | – | 3.89 | 1.96 to 7.71 | 0.000 |
| Every day | – | – | – | 1.94 | 0.83 to 4.50 | 0.124 |
| Constant | 1.76 | | 0.466 | 1.47 | | 0.633 |

Source: Authors' calculation based on data collected for the study.
ref, reference group; SRH, self-rated health.

The association between perceived good quality of marine resources and SRH became stronger (OR=1.77, 95% CI 1.20 to 2.60, p=0.004 in model 3 compared with OR=1.69 in model 1). In other words, controlling for SES variables, the association between perceived environmental quality (especially air quality and marine resources) and SRH became stronger. SES variables thus strengthened the relationship.

Among the SES variables, employment status was the only variable significantly associated with good SRH. Our data show that a large proportion (28.5%) of the respondents were not in the labour market. If bad health is one of the reasons for not joining the labour market, it is not surprising that employed respondents were more likely to report good SRH than non-labour force (OR=1.63, 95% CI 1.05 to 2.53, p=0.029). This likelihood was not different between the unemployed and the non-labour force. In addition, the likelihood of reporting good SRH was not significant for educational attainment and home ownership.

Finally, model 4 shows that HLS measured with frequency of doing physical exercise played the most prominent association with SRH. Doing physical exercise three to five times per week was beneficial for good health (OR=3.89, 95% CI 1.96 to 7.71, p=0.000, the highest OR). However, the health impact of doing physical exercise once or twice, or even every day weekly, was not significantly different from those never doing any exercise. This may indicate that too much exercise has more detrimental effects than benefits to health. However, there is no further information related to physical exercise, such as the duration of exercise, which deserves further research.

Interestingly, the inclusion of physical exercise into the model made the association between perceived air quality and SRH insignificant (p=0.059). In addition, the OR for respondents perceiving fairly clean air became smaller (OR=2.20, model 4), declining from 2.35 (model 3). The association between age and SRH also became insignificant. In other words, physical exercise played a mediating role in the association between perceived environmental quality and good SRH. The inclusion of physical activity strengthened the association between employment status and good SRH, as the OR for employed in model 4 (OR=1.72, 95% CI 1.09 to 2.71, p=0.021) was higher than in model 3.

## DISCUSSION
### Environmental variables
This study contributes to the literature by empirically investigating the relationship between perceived environmental quality and SRH in Brunei, a Southeast Asian country. Our findings suggest the importance of addressing the quality of the natural environment across different dimensions to gauge the differential relationships with SRH. Specifically, the study found that residents who perceived clean air and good marine resources in their surrounding areas were more likely to report good SRH than residents who perceived polluted air and marine resources. Our findings align with a previous study showing that marine quality had a positive relationship with health, which was measured as an objective health.[52] Marine and coastal areas have been extensively polluted with various forms of plastics, such as water bottles, plastic bags, children's toys, wrappers, medical waste and many others. Plastics affect marine resources through the release of microplastics as harmful pollutants.[52] Polluted marine may pollute the fish and other consumable sea commodities such as seaweed, shellfish and others. This may have further implications on physical health as many people consume edible marine resources collected and caught in the sea.

It is therefore important to pay attention to the health impact of marine resources pollution, as the possibility of both onshore and offshore oil spills occurring in oil production areas such as Belait District is very high. The health effect of oil spills could last long and residents' exposure to hydrocarbon has detrimental effects on fetal development and infant mortality, as shown in the case of Nigeria, an oil-producing country.[53] However, land-based activities such as mining, agriculture, forestry, industrial activities, household consumption, river-based transportation and chemical industries can also be a major source of marine pollution.[54]

Furthermore, residents who perceived living in areas with clean air were more likely to report good health. Our findings are consistent with earlier studies.[32 33 55] Poor air quality could cause respiratory infections.[35 56] Polluted air was also associated with cardiovascular disease, lung cancer and lung dysfunction. As part of the Southeast Asian region, Brunei is prone to forest fires, especially during the dry season.[57 58] Seria, one of the selected subdistrict areas, is a hotspot area for forest fires.[57] Bushfires have also become a public health concern in other countries, such as in Australia,[59] Canada[60] and the USA.[61] In addition, exposure to transboundary haze from surrounding ASEAN (Association of Southeast Asian Nations) countries is occasionally experienced in Brunei.[34]

Both indoor and outdoor air pollutions have been a major concern for global public health.[56] Open burning of household rubbish is another source of air pollution. Air pollution, measured objectively by particulate molecule ($PM_{10}$) exceeding the level of good air quality, in Belait had the longest recorded days among districts.[62] A lesson from various studies around the Ecuadorian Amazon regions was that crude oil production activities contributed negative health impact to communities in the form of various cancers, such as cancers of the rectum, skin, kidney, stomach, soft tissue and cervix, as well as leukaemia.[63] Therefore, policy-making needs to consider the surrounding environment to improve people's health.

At the same time, our findings also indicate that commitment to improving air quality was not important in good SRH, although perceived air quality was positively related

to good SRH. Perhaps the respondents believed that the government could do better in preserving air quality.

Noise pollution among the urban residents of China was positively associated with poor SRH.[26 32] However, in the case of Brunei, our findings reveal that perceived olfactory (odour) and noise pollutants were not significantly related to health, although 26.3% of the selected sample perceived themselves to be exposed to olfactory and noise pollutants.

A previous study found that the impact of noise on health was mostly from road traffic.[64] Noise from road traffic may cause nervousness and disturb sleep quality. Nevertheless, as road traffic is relatively low in Brunei, and in Belait in particular, compared with many other countries, noise pollutants may not have any impact on health in our study. Moreover, literature review[65] revealed that there is limited empirical evidence on the association between olfactory pollutants and health. As such, our findings may indicate that olfactory and noise pollution might be a nuisance but does not necessarily affect health.

## Non-environmental variables

Another important finding is that HLS such as frequent physical exercise can compensate for the health impact of perceived environmental quality. Physical activity mediated the relationship between perceived air quality and SRH. Our findings show that if residents actively and regularly do physical exercise, the difference in the prevalence of good SRH became insignificant between residents living in areas with perceived clean air and in areas with polluted air. In other words, regular physical activity can mitigate the adverse environmental consequences on health. Our findings reveal that regular exercise is positively associated with good SRH, confirming earlier studies.[22 41 47]

Previous studies showed that the association between age and health was non-conclusive. Age was not significantly associated with SRH.[26] On the other hand, in our study, age was negatively associated with good SRH when the model was not controlled with physical activity. This means that as people became older, the prevalence of good health declined, supporting earlier studies.[24 32 41 46]

However, our findings further suggest a mediating effect of physical activity on the relationship between age and SRH, which turned this relationship to insignificant. In other words, for the same level of physical activity, age did no longer differentiate SRH. The result indicates that older people who exercised regularly may have an insignificant difference in good health compared with younger people. In other words, physical activity plays a very important role in SRH for all ages.

Another important variable associated with good SRH was employment. Previous studies indicated that work–health relationships can be simultaneous. The relationship can be a result of bias in which only healthy people are willing to work and the unhealthy ones are leaving their jobs. Poor SRH can thus be associated with being unemployed or with an increased risk of job loss.[66] On the other hand, the relationship between employment and health had mixed results; being employed can be associated with good SRH or bad SRH or can have no relationship.[66] Being employed in poorly paid jobs or stressful jobs can be associated with poor SRH. However, our study found a positive association—being employed was more likely to be related to good SRH (OR=1.72, 95% CI 1.09 to 2.71, p=0.021) than being out of the labour force or being unemployed.

## Methodological limitations of the study

Our study has a few limitations. The first limitation is that there is no comparable measure of objective health available at the individual level in the sample. The second limitation is that the questionnaire recorded information on 'olfactory pollutant' and 'noise pollutant' in one group, yet these pollutants may have different health effects. The third limitation is that there was only one variable for social environment. Future study requires separating the questions for these two pollutants and collecting more information that measures the quality of the social environment. The fourth limitation is that this study has not examined the dynamics of the relationship and a longitudinal study to address this issue is needed.

## CONCLUSION

Our study concludes that perceptions of marine resources and air quality mattered to residents' SRH status. Therefore, special attention must be spent to improving air quality and marine resources. Moreover, our findings show that physical exercise and employment play important roles in health. There is a strong need for policy interventions to mitigate the negative health impact of environmental quality by promoting physical exercise and employment. Further studies should explore more comprehensive measurements of environmental quality, HLS and employment variables to assess health and environmental quality.

**Acknowledgements** The authors would like to thank Brunei Shell Petroleum (BSP) Sendirian Berhad for providing the funding to conduct this study. The authors would also like to thank Professor Gour Dasvarma for valuable input and constructive comments. They also thank the field research team who worked hard to complete the data collection.

**Contributors** ENA designed the concept, analysed the data, chose the method and wrote the manuscript. C-YH supervised the project administration and data collection, and wrote, reviewed and edited the final draft. LS coordinated the data collection and reviewed and edited the final draft. AT facilitated the funding acquisition, analysed the data, reviewed the methods and edited the final draft. All authors contributed to reading and editing and approved the final draft of the manuscript. ENA was the study guarantor.

**Funding** The work was supported by a research grant sponsored by the Brunei Shell Petroleum (BSP) Sendirian Berhad (reference no: BSP/SI2018.011/CEA4).

**Competing interests** None declared.

**Patient and public involvement** Patients and/or the public were not involved in the design, or conduct, or reporting, or dissemination plans of this research.

**Patient consent for publication** Not required.

**Ethics approval** This study involves human participants and was approved by the University Research Ethics Committee at Universiti Brunei Darussalam (reference

number UBD/OAVCR/UREC/Sep19-04). Participants gave informed consent to participate in the study before taking part.

**Provenance and peer review** Not commissioned; externally peer reviewed.

**Data availability statement** No data are available to the public due to data agreement.

**ORCID iD**
Evi Nurvidya Arifin http://orcid.org/0000-0002-7852-1063

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
