## [Reviewer comments · BMJ Open]

ARTICLE DETAILS

TITLE (PROVISIONAL)	Self-rated Health and Perceived Environmental Quality in Brunei Darussalam:A cross-sectional study
AUTHORS	Arifin, Evi Nurvidya; Hoon, Chang-Yau; Slesman, Ly; Tan, Abby

VERSION 1 – REVIEW

REVIEWER	Dasvarma, Gour Flinders University, College of Humanities, Arts and Social Sciences
REVIEW RETURNED	20-Apr-2022

GENERAL COMMENTS	This manuscript is about the influence of a combination of a set of realities and perceptions on individuals' perception of their health status. The discussions, conclusions and policy recommendations though, mostly centre on the influence of perceptions of natural and social environments on an individual's perceived health status. Self-perceived health status has been used in various studies (Paul et.al, 2015; Xiao and Barber, 2008). However, what this reviewer finds unusual is that this research has used perceived status of natural and social environments as predictors of perceived health status. The authors' justification for using perceived status of the environment is that objective measures of the environment based on actual data do not exist at the individual level. There is a difference between perception and reality. The authors should try to validate the respondents' rating of their natural and social environment with data on air and water pollution and data on crime statistics. Such data, as everyone knows are measured at aggregate levels, but they are applicable to all individuals living within that aggregate space. The manuscript is fairly well-written, but there is a need for some editing for English expressions, particularly in the use of prepositions. The manuscript may be considered for publication provided the authors address the comments including those via sticky note copied below. I have attached a copy of the manuscript with my sticky note comments. Comments via sticky note: 1. Objectives (Abstract). Line 44. Please correct the preposition. perception on to perception of.
---

	 2. Abstract. Objectives. Line 45. "...accounting for...". Do you mean "...after controlling for... "? 3. Abstract. Setting, lines 48-49. "It was conducted ...", please write "The study was conducted in ...". Please also state why Belait district was chosen. 4. Abstract. Setting, lines 49-50. Please give the dates. 5. Abstract. Participants: Line 52. Who were these 1,000 respondents? What was the total population of persons aged 18 years and over in Belait district? 6. Abstract. Outcome measures, lines 54-55. Change "Perception on." to "Perception of". 7. Abstract. Outcome measures. lines 54-55. add "perceptions of natural environment..." 8. Abstract, Outcome measures. line 57. Change "model" to "models". 9. Abstract. Results. There is no need to write "p-value" Just write p=... 10. Abstract. Conclusions. Please compare SRH with some objective measure of health, if possible. Otherwise this will be a weakness of the study. 11. Article Summary. Lines 28-29. self-rated health, a subjective measure. 12. Article Summary. Lines 32-33. How is using cross-sectional survey a strength? 13. Article Summary. Lines 41-42. Another weakness is that there is no comparable measure of objective health. 14. But the greatest weakness, in the opinion of this reviewer is that there is an implicit bias in terms of subjective rating of health as well as the factors that are hypothesised to affect self-rated health. Those respondents who rate their natural and social environment as good would most likely also rate their health status as being good. Otherwise, they would see themselves as contradicting themselves. The authors should try to validate the respondents' rating of their natural and social environment with data on air and water pollution and data on crime statistics. Such data, as everyone knows are measured at aggregate levels, but they are applicable to all individuals living within that aggregate space. 15. Introduction. Lines 5-6. Please give a reference. 16. Introduction. Lines 5-6. Which SDG? 17. Introduction. Line 18. Change "happen" to "happens". 18. Introduction. Line 23. give references. 19. Introduction. Lines 25-26. Delete or modify the first sentence. "The communicable diseases, for example, may spread to the community". That is why they are called communicable diseases. 20. Introduction. Line 28. "...disruption on..." change to "...disruption of...". 21. Introduction. Line 38. Give reference(s). 22. Introduction. Lines 57-58. Please elaborate in brief how democracy is associated with health. 23. Page 5 of 42. Lines 11-13. Objective measures of environmental quality are constants. They are applicable to all individuals living in the same environment. It is the subjective measures of
--	---

environmental quality that vary from one individual to another. The researcher should find out why subjective measures of environmental quality vary between individuals. Therefore, this research should have compared objective measures of, say air quality, or water quality with those as perceived by individuals (subjective measures). Please address this comment as best you can.

24. Page 5 of 42. Lines 18-19. When you say, "health status" you should say "self-rated health status."
25. Page 5 of 42, lines 38-43. Self-rated health has been used also as a part of multi-dimensional human wellbeing in Assam, India (https://www.niti.gov.in/writereaddata/files/human-development/Assam_HDR_30Sep2016.pdf); in Timor-Leste (<https://hdr.undp.org/en/content/national-human-development-report-2018-timor-leste>); and in and Australia (<https://doi.org/10.1371/journal.pone.0252898>). In all these studies, individuals have rated their physical and psychological health based on 5 or 6 questions. Please state, in the present study under review, how many questions were asked of the survey respondents to rate their health status.
26. Page 5 of 42. Line45-46. See the previous comment.
27. **Page 7 of 42, lines 28-38. Please explain why it is important to examine the relationship between natural and social environmental qualities and SRH.**
28. Page 8 of 42. Lines 39-40. Explain why the sample size was taken as 1,000. What was the total population aged 18 years and above in Belait district?
29. Page 10 of 42. Lines 20-23. "Natural environmental quality were collected...". Do you mean "Data on natural environmental quality were collected..."?
30. Page 12 of 42. Lines 8-11. With perceptions of water quality, dwelling quality and social environment quality, a score of 1 is assigned to a negative perception, whereas, for all other variables, the score of 1 is assigned to a positive perception. This might create problem in interpreting the results of logistic regression. I hope you are aware of that and interpreted the findings accordingly.
31. Page 11 of 42. Lines 40-41. "1 for male and 0 for otherwise". Please change "otherwise" to "female", unless there was an option for other sexual orientations than male and female.
32. Page 11 of 42. Lines 47-48. Does the marital status "single" include never married, widowed and widower? Or is it just "never married"?
33. Page 11 of 42. Lines 50-53. Please state the difference between (Brunei) Malay and Indigenous Malay.
34. Page 13 of 42. Lines 57-59. Please give a reference to the source of the logistic regression equation model.
35. Page 13 of 42. Lines 3-11. Not everyone can access the study you have referred to (Reference 10 in your manuscript). Therefore, please explain why the model was applied in four stages. Why did you not have only one model with all the control variables and show simple ORs and Adjusted ORs?
36. Page 13 of 42. Lines40-41. (48.5%). This should be 58.5%.

	37. Tables. Please give the sources of all tables. Even if they are drawn from your own data, cite the reference as "Computed by the authors based on data collected for this study", or some reference to that effect. 38. Page 13 of 42. Lines 55-58. Please re-write this sentence. "Most residents" cannot possibly perceive good air quality, very good quality or air pollution, all at the same time. 39. Page 14 of 42. Lines 22-24 (Table 2). Please show the Chi-square values (for strength of association) as well as their p-values (for statistical significance). 40. Page 14 of 42. Lines 44-47. How do you explain this anomaly? 41. Page 14 of 42. Lines 54-55. This p-value is 0.001 (not 0.010). 42. Page 14 of 42. Last three lines. The prevalence is NOT equal - it is 74% and 68.9% respectively, although the chi-square is statistically not significant. 43. Page 15 of 42. Lines 36-41. Please show the chi-square values so that one can see the strength of association between variables. p-values do not show the strength of association; they merely show how likely it is for the association found in the sample to be true in the population from which the sample has been drawn. 44. Page 15 of 42. Section heading: Logistic regression of self-rated health. Why have all the predictor variables in the bivariate (chi-square) analysis, regardless of the p-values been included in the logistic regression models? Why not include just those variables which have p-values of less than 0.05 in the bivariate analysis? 45. Page 16 of 42. Lines 6-8. The prevalence in the chi-square test you have referred to are 77.4% and 73.9 % respectively (Table 2). Most probably these percentages are not statistically significantly different from each other. 46. Page 16 of 42. Lines 38-40. "A social environmental condition measured by perception on crime was neither significantly associated with SRH". Please re-write the sentence. (Do you mean "Neither was the social environmental condition measured by perception on crime significantly associated with SRH".?) 47. Page 16 of 42. Lines 52-53. The OR is almost 1.00 (It is 0.98), which indicates little or no influence of age on SRH. 48. Page 17 of 42. Lines 3-4. "Men and women had the same likelihood of reporting good SRH". But the OR is statistically not significant. 49. Page 17 of 42. Lines 3-5. Do you have any comment about ethnic group and SRH? 50. Page 17 of 42. Lines 48-51. A large proportion (28.5%) of the respondents are not in the labour force. This is surprising considering the district you have studied (Belait) is a gas and oil rich district. However, one of the reasons for not being in the labour force is bad health. Therefore, it may not be surprising to find that those who are in the labour force are much more likely to state good SRH. 51. Page 18 of 42. Lines 13-16. 52. "However, the health impact of doing physical exercise once or twice, or even everyday weekly was not significantly
--	---

	different from those never doing any exercise". Why do you think this was so? 53. Page 18 of 42. Lines 22-23. 54. p-values show the level of statistical significance. They do not show the strength of association. 55. Page 18 of 42. Lines 43-48. What does a country's high per capita GDP got to do with the association between perceived environmental quality and SRH? Please remove any reference to GDP if this has no role in the associations you have discussed in this manuscript. 56. Page 19 of 42. Lines 3-6. Are you referring to self-rated health or an objective measure of health? 57. Page 21 of 42. Line 3. Reference to noise pollution from road traffic. Is this evident from the current study or is it a general statement? If it is the latter, then please re-write the sentence. References. Paul, P., Hakobyan, M. & Valtonen, H. 2015. The association between self-perceived health status and satisfaction with healthcare services: Evidence from Armenia. BMC Health Serv Res 16, 67 (2016). https://doi.org/10.1186/s12913-016-1309-6. Xiao H, Barber JP. The effect of perceived health status on patient satisfaction. Value Health. 2008 Jul-Aug;11(4):719-25. doi: 10.1111/j.1524-4733.2007.00294.x. Epub 2007 Dec 17. PMID: 18179667.
--	--

VERSION 1 – AUTHOR RESPONSE

“Self-rated Health and Perceived Environmental Quality in Brunei Darussalam: A cross-sectional study”.

Submitted to *BMJ Open*

General comments

This manuscript is about the influence of a combination of a set of realities and perceptions on individuals' perception of their health status. The discussions, conclusions and policy recommendations though, mostly centre on the influence of perceptions of natural and social environments on an individual's perceived health status. Self-perceived health status has been used in various studies (Paul et.al, 2015; Xiao and Barber, 2008). However, what this reviewer finds unusual is that this research has used perceived status of natural and social environments as predictors of perceived health status. The authors' justification for using perceived status of the environment is that objective measures of the environment based on actual data do not exist at the individual level. There is a difference between perception and reality. The authors should try to validate the respondents' rating of their natural and social environment with data on air and water pollution and data on crime statistics. Such data, as everyone knows are measured at aggregate levels, but they are applicable to all individuals living within that aggregate space.

The manuscript is fairly well-written, but there is a need for some editing for English expressions, particularly in the use of prepositions.

The manuscript may be considered for publication provided the authors address the comments including those via sticky note copied below. I have attached a copy of the manuscript with my sticky note comments.

Comments via sticky note:

1. Objectives (Abstract). Line 44. Please correct the preposition. perception on to perception of.
Corrected
2. Abstract. Objectives. Line 45. "...accounting for...". Do you mean "...after controlling for..."?
Corrected
3. Abstract. Setting, lines 48-49. "It was conducted ...", please write "The study was conducted in ...". Please also state why Belait district was chosen.
Corrected
4. Abstract. Setting, lines 49-50. Please give the dates.
The survey was conducted between 17 October to 11 November 2019
5. Abstract. Participants: Line 52. Who were these 1,000 respondents? What was the total population of persons aged 18 years and over in Belait district?

Added and explained more under Data section.
6. Abstract. Outcome measures, lines 54-55. Change "Perception on." to "Perception of".
Corrected
7. Abstract. Outcome measures. lines 54-55. add "perceptions of natural environment..."
Added
8. Abstract, Outcome measures. line 57. Change "model" to "models".
Changed
9. Abstract. Results. There is no need to write "p-value" Just write p=...
Revised. It's revised throughout the text.
10. Abstract. Conclusions. Please compare SRH with some objective measure of health, if possible. Otherwise this will be a weakness of the study.

There is no comparable objective measure of health
11. **Article Summary.** Lines 28-29. self-rated health, a subjective measure.
Revised
12. Article Summary. Lines 32-33. How is using cross-sectional survey a strength?

The sentence was revised. Another strength is that it used a representative and relatively large sample of a district.
13. Article Summary. Lines 41-42. Another weakness is that there is no comparable measure of objective health.
Revised and added

14. **But the greatest weakness, in the opinion of this reviewer is that there is an implicit bias in terms of subjective rating of health as well as the factors that are hypothesised to affect self-rated health. Those respondents who rate their natural and social environment as good would most likely also rate their health status as being good. Otherwise, they would see themselves as contradicting themselves. The authors should try to validate the respondents' rating of their natural and social environment with data on air and water pollution and data on crime statistics. Such data, as everyone knows are measured at aggregate levels, but they are applicable to all individuals living within that aggregate space.**

Comments: Earlier studies on the relationship between perceived environmental quality and SRH provide mixed findings. If measures on environmental quality are applicable to all individuals living in the same area. Then there will be no variation among individuals. Thus, there won't be any relationship between environment and health in a specific area. However, individuals' perception of their environmental quality may say more about a variation in the perceptions of environment and health. This study contributes to new empirical findings.

15. Introduction. Lines 5-6. Please give a reference.

Added a reference: United Nation, 2015. Resolution Adopted by the General Assembly on 25 September 2015. Transforming Our World: the 2030 Agenda for Sustainable Development. Geneva: United Nations.

16. Introduction. Lines 5-6. Which SDG?

Revised; Goal 3

17. Introduction. Line 18. Change "happen" to "happens".

Revised

18. Introduction. Line 23. give references.

Added 3 references: Buddlemeyer and Cai 2009; Clarke and Erreygers, 2019; Ananta et al. 2021

19. Introduction. Lines 25-26. Delete or modify the first sentence. "The communicable diseases, for example, may spread to the community". That is why they are called communicable diseases.

Deleted

20. Introduction. Line 28. "...disruption on..." change to "...disruption of...".

Revised

21. Introduction. Line 38. Give reference(s).

Added 3 references: Olivia et al, 2020; Dang et al. 2021; Whitehead et al. 2021

22. Introduction. Lines 57-58. Please elaborate in brief how democracy is associated with health.

Revised

23. **Page 5 of 42. Lines 11-13. Objective measures of environmental quality are constants. They are applicable to all individuals living in the same environment. It is the subjective measures of environmental quality that vary from one individual to another. The researcher should find out why subjective measures of environmental quality vary between individuals. Therefore, this research should have compared objective measures of, say air quality, or water quality with those as perceived by individuals (subjective measures). Please address this comment as best you can.**

Because it is not easy to measure objective measures of environmental quality at the individual level, this paper utilized its proxy -- perceived environmental quality. *Added this into the text. Most importantly, objective measures of environmental quality are applicable to all individuals living in the same environment. It is, therefore, no variation as compared to the subjective measures of environmental quality that can vary from one individual to another.*

24. Page 5 of 42. Lines 18-19. When you say, "health status" you should say "self-rated health status."

Revised as suggested

25. Page 5 of 42, lines 38-43. Self-rated health has been used also as a part of multi-dimensional human wellbeing in Assam, India (https://www.niti.gov.in/writereaddata/files/human-development/Assam_HDR_30Sep2016.pdf); in Timor-Leste (<https://hdr.undp.org/en/content/national-human-development-report-2018-timor-leste>); and in Australia (<https://doi.org/10.1371/journal.pone.0252898>).

In all these studies, individuals have rated their physical and psychological health based on 5 or 6 questions. Please state, in the present study under review, how many questions were asked of the survey respondents to rate their health status.

Thank you very much for sharing the references. These references have been included to the text. More explanations have also been added under the method section.

26. Page 5 of 42. Line45-46. See the previous comment.

Timor Leste, one of Southeast Asian countries, has been mentioned and cited in the earlier part of this paragraph

- 27. Page 7 of 42, lines 28-38. Please explain why it is important to examine the relationship between natural and social environmental qualities and SRH.**

Added more explanation

28. Page 8 of 42. Lines 39-40. Explain why the sample size was taken as 1,000. What was the total population aged 18 years and above in Belait district?

This survey interviewed 1,000 randomly selected respondents aged 18 years and above. The population aged 18 years and above in Belait was roughly about 72% based on the 2016 data (DEPS, 2018). The 1000 selected respondents were approximately about 2% of the population aged 20 years and above. Unfortunately, the age composition of Belait's population in 2019 was unavailable. This size was randomly drawn with a 95% confidence level and $\pm 3\%$ margin of error.

29. Page 10 of 42. Lines 20-23. "Natural environmental quality were collected...". Do you mean "Data on natural environmental quality were collected..."?

Revised as suggested

30. Page 12 of 42. Lines 8-11. With perceptions of water quality, dwelling quality and social environment quality, a score of 1 is assigned to a negative perception, whereas, for all other variables, the score of 1 is assigned to a positive perception. This might create problem in interpreting the results of logistic regression. I hope you are aware of that and interpreted the findings accordingly.

Yes, we are aware of this. We don't use zero and one in the Tables, instead put the labels.

31. Page 11 of 42. Lines 40-41. "1 for male and 0 for otherwise". Please change "otherwise" to "female", unless there was an option for other sexual orientations than male and female.
Thank you. We changed it to female.

32. Page 11 of 42. Lines 47-48. Does the marital status "single" include never married, widowed and widower? Or is it just "never married"?

We revised the marital status as married, ever married (divorced and widowed) and single/never married.

33. Page 11 of 42. Lines 50-53. Please state the difference between (Brunei) Malay and Indigenous Malay.

Revised as: Ethnicity consisted of four groups: Brunei Malay, Indigenous Malay, Chinese and Others. As of the Brunei Nationality Act 1961, the Malay group consists of seven ethnic groups: Brunei Malay or Brunei, Belait, Bisaya, Dusun, Kedayan, Murut and Tutong (DEPS, 2021). Brunei Malay is culturally Malay and religiously Islam which is similar to the Malay in West Malaysia, while other indigenous Malays are culturally and religiously diverse (Loo 2009).

34. Page 13 of 42. Lines 57-59. Please give a reference to the source of the logistic regression equation model.

Added a reference: Hosmer, Lemeshow and Sturdivant (2013).

35. Page 13 of 42. Lines 3-11. Not everyone can access the study you have referred to (Reference 10 in your manuscript). Therefore, please explain why the model was applied in four stages. Why did you not have only one model with all the control variables and show simple ORs and Adjusted ORs?

Following previous study (Park & Lee, 2013), the model was applied in few stages to examine each group of variables at each stage. This paper has four stages to observe the possible changes played by perceived environmental quality, demographic variables, socio-economic variables and lifestyle variables on SRH.

36. Page 13 of 42. Lines 40-41. (48.5%). This should be 58.5%.

Thank you very much, corrected it as suggested

37. Tables. Please give the sources of all tables. Even if they are drawn from your own data. cite the reference as "Computed by the authors based on data collected for this study", or some reference to that effect.

Added sources as suggested in all tables.

38. Page 13 of 42. Lines 55-58. Please re-write this sentence. "Most residents" cannot possibly perceive good air quality, very good quality or air pollution, all at the same time.

Revised: The analysis of the survey data shows a good perception of air quality in the area the respondents reside. The descriptive statistics show that 80.2% of respondents perceived they reside in fairly good air quality areas. Only 12.5% of respondents perceived they live in very good air quality areas. In other words, only a smaller percentage of respondents (7.3%) perceived they live in air-polluted areas.

39. Page 14 of 42. Lines 22-24 (Table 2). Please show the Chi-square values (for strength of association) as well as their p-values (for statistical significance).

The Chi-square values have been added to Tables 1 and 2.

40. Page 14 of 42. Lines 44-47. How do you explain this anomaly?

Sentence revised. In other words, commitment to improving air quality was not associated with good SRH, though perceived air quality was positively related to good SRH. This indicates the existence of a knowledge-attitude gap.

41. Page 14 of 42. Lines 54-55. This p-value is 0.001 (not 0.010).

Thank you very much, corrected it as suggested

42. Page 14 of 42. Last three lines. The prevalence is NOT equal - it is 74% and 68.9% respectively, although the chi-square is statistically not significant.

Revised

- 43. Page 15 of 42. Lines 36-41. Please show the chi-square values so that one can see the strength of association between variables. p-values do not show the strength of association; they merely show how likely it is for the association found in the sample to be true in the population from which the sample has been drawn.**

The Chi-square values have been added to Tables 1 and 2.

44. Page 15 of 42. Section heading: **Logistic regression of self-rated health.**

Why have all the predictor variables in the bivariate (chi-square) analysis, regardless of the p-values been included in the logistic regression models? Why not include just those variables which have p-values of less than 0.05 in the bivariate analysis?

Comments: We are aware of what you mentioned about the insignificant variables. However, we are in the position that including all variables into the model, regardless of their significance, means they are part of the system. Their presence affects the magnitude of the coefficient.

45. Page 16 of 42. Lines 6-8. The prevalence in the chi-square test you have referred to are 77.4% and 73.9 % respectively (Table 2). Most probably these percentages are not statistically significantly different from each other.

Yes, agree. We tested that these percentages are not statistically significantly different from each other.

46. Page 16 of 42. Lines 38-40. "A social environmental condition measured by perception on crime was neither significantly associated with SRH". Please re-write the sentence. (Do you mean "Neither was the social environmental condition measured by perception on crime significantly associated with SRH".?)

Revised as suggested

47. Page 16 of 42. Lines 52-53. The OR is almost 1.00 (It is 0.98), which indicates little or no influence of age on SRH.

The change is small but significant.

48. Page 17 of 42. Lines 3-4. "Men and women had the same likelihood of reporting good SRH". But the OR is statistically not significant.

Because the OR is insignificant, there is no difference between men and women on good SRH.

49. Page 17 of 42. Lines 3-5. Do you have any comment about ethnic group and SRH?
This deserves further study. Yet, a study in Singapore (Lim et al. 2007) found that ethnicity has a positive relationship with good SRH in which Malay tend to have a higher likelihood of good SRH than the Chinese.
50. Page 17 of 42. Lines 48-51. A large proportion (28.5%) of the respondents are not in the labour force. This is surprising considering the district you have studied (Belait) is a gas and oil rich district. However, one of the reasons for not being in the labour force is bad health. Therefore, it may not be surprising to find that those who are in the labour force are much more likely to state good SRH.
Agree and revised.
51. Page 18 of 42. Lines13-16. "However, the health impact of doing physical exercise once or twice, or even everyday weekly was not significantly different from those never doing any exercise". Why do you think this was so?
This may indicate that too much exercise has more detrimental effects than benefits on health. However, there is no further information related to physical exercise such as the duration of exercise. This deserves further research.
52. Page 18 of 42. Lines 22-23. p-values show the level of statistical significance. They do not show the strength of association.

Revised.
53. Page 18 of 42. Lines 43-48. What does a country's high per capita GDP got to do with the association between perceived environmental quality and SRH? Please remove any reference to GDP if this has no role in the associations you have discussed in this manuscript.
Removed as suggested
54. Page 19 of 42. Lines3-6. Are you referring to self-rated health or an objective measure of health?

Revised. It's referring to an objective measure of health.
55. Page 21 of 42. Line 3. Reference to noise pollution from road traffic. Is this evident from the current study or is it a general statement? If it is the latter, then please re-write the sentence.
Revised

References.

Paul, P., Hakobyan, M. & Valtonen, H. 2015. The association between self-perceived health status and satisfaction with healthcare services: Evidence from Armenia. *BMC Health Serv Res* **16**, 67 (2016). <https://doi.org/10.1186/s12913-016-1309-6>.

Xiao H, Barber JP. The effect of perceived health status on patient satisfaction. *Value Health*. 2008 Jul-Aug;11(4):719-25. doi: 10.1111/j.1524-4733.2007.00294.x. Epub 2007 Dec 17. PMID: 18179667.

VERSION 2 – REVIEW

REVIEWER	Dasvarma, Gour Flinders University, College of Humanities, Arts and Social Sciences
REVIEW RETURNED	05-Jun-2022

GENERAL COMMENTS	No further comments.
----------------------